# Informal Employment, Working Conditions, and Self-Perceived Health in 3098 Peruvian Urban Workers

**DOI:** 10.3390/ijerph19106105

**Published:** 2022-05-17

**Authors:** Michael Silva-Peñaherrera, Amaya Ayala-Garcia, Erika Alferez Mayer, Iselle Sabastizagal-Vela, Fernando G. Benavides

**Affiliations:** 1Center for Research in Occupational Health (CiSAL), University Pompeu Fabra, Edificio PRBB, Doctor Aiguader, 88, 08003 Barcelona, Spain; amaya.ayala@upf.edu (A.A.-G.); ealferezm@gmail.com (E.A.M.); fernando.benavides@upf.edu (F.G.B.); 2Hospital del Mar Research Institute (IMIM), Parc Salut Mar, 08003 Barcelona, Spain; 3Centro de Investigación Biomédica en Red (CIBER) of Epidemiology and Public Health (CIBERESP), 28029 Madrid, Spain; 4Instituto Nacional de Salud, Lima 15072, Peru; isabastizagalv@gmail.com; 5Unidad de Medicina Ocupacional y Medio Ambiente, Facultad de Medicina Alberto Hurtado, Universidad Peruana Cayetano Heredia, Lima 15102, Peru

**Keywords:** informality, working conditions, self-reported health, survey

## Abstract

Peru has one of the highest informal employment rates in Latin America (73%). Previous studies have shown a higher prevalence of poor self-perceived health (P-SPH) in informal than in formal workers. The aim of this study was to analyze the role of working conditions in the association between informality and SPH in an urban working population in Peru. We conducted a cross-sectional study based on 3098 workers participating in the working conditions survey of Peru 2017. The prevalence of P-SPH and exposure to poor working conditions were calculated separately for formal and informal employment and were stratified by sex. Poisson regression models were used to assess the association between P-SPH and informal employment, with crude and adjusted prevalence ratios (PR) for working conditions. Informal employment affected 76% of women and 66% of men. Informal workers reported higher exposition to poor working conditions than formal workers and reported worse SPH. Informal workers had a higher risk of P-SPH than formal workers: PR 1.38 [95% CI: 1.16–1.64] in women and PR 1.27 [95% CI: 1.08–1.49] in men. Adjustment by working conditions weakened the association in both sexes. In women, this association was only partially explained by worse working conditions; PR 1.23 [95% CI: 1.04–1.46]. Although some of the negative effect of informal employment on workers´ health can be explained by the characteristics of informality per se, such as poverty, a substantial part of this effect can be explained by poor working conditions.

## 1. Introduction

Informal employment is probably the most precarious type of employment and is also one of the most widespread, affecting more than 60% of workers worldwide [1]. It is an important social determinant of health that represents a serious public health problem [2]. Informal workers are not recognized or protected by legal and regulatory frameworks, and they have no social protection or any power to negotiate working or employment conditions. The International Labour Organization (ILO) defines informal employment as the total number of informal jobs, whether in formal sector enterprises, in informal sector enterprises, or as domestic workers paid by households. Informal jobs are those whose employment relationship is not subject to national labor legislation, income taxation, social protection or entitlement to employment benefits [3] According to the World Bank, this high percentage of informality could delay the recovery of the economy after the COVID-19 pandemic [4]. A recent study shows that excess mortality in the working age population during the pandemic in Peru was higher than in most countries in the region [5]. The pandemic is likely to have increased, at least temporarily, the informal employment rates, and to have resulted in high income losses to this working population. As a response to the increasingly unprotected situation of working people, particularly in middle- and low-income countries, the ILO has included among its priorities the formalization of the informal economy [6], and the UN’s global agenda 2030 for sustainable development has included the objective of decent work for all in its eighth goal [7]. These objectives will require significant efforts and the implementation of multiple strategies to facilitate the transition to formality [8] In Peru, informal employment involves 73% of the working population [9], well above of the Latin American average of 53%. Therefore, the informal economy plays an important role in the dynamics of the Peruvian economy, generating around one-third of its gross domestic product [4].

The literature on the impact of informal employment on health is scarce. Previous studies have found that high rates of informal employment were associated with poor mental and poor self-perceived health (P-SPH) [10,11]. A possible hypothesis is that informal employees are more likely to be exposed to poor working conditions and precariousness, which are related to occupational diseases, injuries and disabilities that reduce their work capacity and earning potential [12]. In addition, more and more women are entering the labor market, but a large proportion continue to be employed in more precarious jobs, with lower wages, high levels of informality, and job insecurity. This, together with women’s unequal participation in unpaid work, can have a detrimental effect on their general and mental health [13]. However, the role of working conditions in the association between SPH and informality and its differential effect on men and women remains unclear. The objective of this study was to assess the role of working conditions on the relationship between informal employment and health status in urban workers in Peru.

## 2. Materials and Methods

This cross-sectional study was based on data from 3098 urban workers participating in the first working conditions survey of Peru conducted between November 2016 and June 2017—the most up-to-date data available on working conditions and employment in Peru. The sampling was probabilistic area-based, stratified and multistage and the resulting sample was representative of people older than 14 years old who had worked at least 1 hour the week prior to the survey or who were temporarily absent from work due to vacation, illness or leave, which was according to the ILO definition of employee [14]. Sample selection was multistage probabilistic with a confidence level of 95%. The calculated sample was 3120 people distributed in 520 clusters. Survey-weighting factors were applied for all calculations. Informal workers of both sexes were younger than formal workers (Appendix A). The questionnaire was validated, and its methodology was developed taking the European working conditions survey as a reference [15]. It was administered in a face-to-face interview at the worker´s dwelling. Agricultural and military workers were excluded. A detailed description of the survey methodology is available elsewhere [16].

The main explanatory variable was informal employment. We defined informal employment as the lack of a contract and social security coverage. In Peru, as in many countries, legally recognized workers must be affiliated to a pension system. Thus, not being affiliated is a close proxy for informal employment. We considered informal workers those who responded negatively to the item: Currently, *do you have a discount, contribute, are you affiliated or registered in a retirement system (ONP/AFP)?* The outcome (measure¿) of our study was self-perceived health (SPH), which, as is usual in most epidemiological studies [17], was dichotomized into good SPH in workers reporting good and very good SPH, and P-SPH in those reporting fair, poor, and very poor health in response to the item: *How do you consider your health status has been in the last two weeks?*

Secondary explanatory variables were exposure to the following working conditions items: safety dimension: falls on the same level, falls from a different level, and exposure to machines or tools; ergonomic dimension: awkward postures, load lifting and repetitive movements; hygiene dimension: noise, chemical risk, breathing in dust and fumes, biological risk and radiation; and psychosocial dimension: high work rate, low work control, hiding emotions, not applying knowledge, not learning, high workload, lack of supervisor support, lack of coworker support, lack of recognition (Appendix A). The exposure level was measured by a general item: *In your workplace, how often are you exposed to…?* and the possible responses on a five-category Likert scale were dichotomized for descriptive analyses into poor working conditions for those who answered “always, often, and sometimes”; and good working conditions for those reporting “rarely” and “never” (Table 1 and Table 2). This is the most common way to analyze exposure to poor working conditions in a description analysis [18]. For regression analysis, exposure to working conditions was treated as a continuous variable (Figure 1). The possible responses in which each category represented a numerical value on the Likert scale (always = 5, often = 4, sometimes = 3, rarely = 2 and never = 1, except for positive psychosocial conditions, which were scored the other way around) were summed in an individual score for each working condition dimension (safety, hygiene, ergonomic, and psychosocial dimensions). The higher the score, the higher the exposure level (Appendix A). Finally, we considered age (≤24, 25–44, 45–64, and ≥65 years) as a covariate in the association model. All analyses were stratified by sex and weighted by age groups, sex, and industry sector (primary, secondary, and tertiary).

In the analysis, first we estimated prevalence of P-SPH by formal and informal employment, and calculated the prevalence of exposure to poor working conditions as a dichotomous variable in people reporting P-SPH. The chi-square test was calculated to assess differences in the distribution between comparison groups. Second, Poisson regression models were used to measure the association between P-SPH and informal employment, taking formal employment as the reference group. This regression model was selected over other regression models because it provides unbiased estimates [19,20]. However, the prevalence ratio (PR) should be interpreted cautiously due to the cross-sectional nature of the study, which could induce a reverse causality bias. PR and their 95%CI, both crude and adjusted by age and working conditions, were estimated as a continuous variable. The analysis was conducted with Stata v.13.

## 3. Results

Overall, 71% of the sample were working in informal employment (77% of women and 66% of men). In informal employees, the mean age was 38 years in men and 37 years in women. In formal employees, the mean age was 41 years in men and 38 years in women. Informal workers showed higher P-SPH than formal workers, especially women (47.3%) compared with men (37.6%) (Table 1).

Informal workers reported a higher prevalence of poor safety, hygiene and ergonomic working conditions than workers in formal employment, especially men. The highest prevalence of poor working conditions was found in informal employees in relation to repetitive movements (71.7% in men and 62.6% in women), followed by load lifting among men (63.1%). The prevalence of poor psychosocial conditions differed from that of other working conditions. Both men and women with formal jobs showed a high prevalence of low control at work (64.9% of men, 69.8% of women) and hiding emotions (50.0% of men, 62.7% of women) (Table 1).

Workers reporting P-SPH had high levels of poor safety, hygiene and ergonomic conditions (Table 2). Women with P-SPH consistently reported a higher prevalence of exposure than men. Women reported the highest percentage of exposure to machine tools (58.9%) and falls from a different level (56.3%). Men followed the same trends with lower estimates, with the highest value being exposure to chemical risk (44.7%).

When we examined the association between P-SPH and informal employment (Figure 1), both women and men in informal employment showed a higher probability of reporting P-SPH than those in formal employment (1.27 [95% CI: 1.08–1.49] men and 1.38 [95% CI: 1.16–1.64] women). When the analysis was adjusted by age, the association became stronger in both sexes (1.35 [95% CI: 1.15–1.58] in men and 1.41 [95% CI: 1.18–1.67] in women). When adjusted individually for each working condition score, this association held among women and men. When the analysis was adjusted by all working conditions and age, the association lost strength, especially in men (1.17 [95% CI: 1.0–1.38]).

## 4. Discussion

Our results show that workers in informal employment had a statistically significantly higher prevalence of P-SPH than those in formal employment, and that this association was stronger in women. When adjusted by working conditions, the association weakened and almost disappeared in men but remained in women. These results suggest that the observed negative effect on informal workers’ health is partially explained by poor working conditions. However, some of the effect could be explained by informality per se, which could be related to other mechanisms such as income insecurity, poverty, precarious living conditions, lack of access to health services and the absence of social protection benefits. These social and economic factors, together with living conditions, could play an important role in this association. This is in line with previous literature showing the protective role of social protection on health [21,22].

The higher prevalence of P-SPH among workers in informal employment is consistent with previous studies in the region. For example, a study performed in Argentina showed that informal workers had a higher prevalence of P-SPH and low psychophysical well-being, as well as a higher prevalence of poor working conditions (hygiene and ergonomic) than formal workers [23]. A study in Brazil revealed that informal part-time workers reported arthritis, bronchitis, heart disease, cirrhosis, depression, and chronic disease more often than their counterparts in full-time work with social protection [24]. Another study in 15 Latin American countries found that manual (skilled and non-skilled) jobs had the highest P-SPH, and that around 42% of P-SPH in men and 31% in women could be avoided if they had the working and employment conditions of workers in non-manual skilled jobs [25]. Due to lack of government control, working conditions in informal employment frequently breach health and safety laws, often to avoid the costs associated with the regulatory requirements (safety training or protective equipment) or to pay lower salaries. As a result, health determinants such as education and income are related to informal employment and occupational hazards are more common [26].

Men had worse working conditions than women, but women had a higher prevalence of P-SPH than men and showed a stronger association between informality and P-SPH. A study on the prevalence of P-SPH among men and women that did not include employment status (formal or informal) concluded that this prevalence was consistently higher in women than in men [27]. Our results suggest that P-SPH associated with informal employment could have a two-fold explanation, although this effect differs slightly by sex. While working conditions seem to have a stronger impact on men’s health, probably because they have riskier jobs, other determinants could be operating in women, such as the burden of the combination of paid and unpaid reproductive work, which could affect the health of women in informal employment [28]. These findings are consistent with those of other studies conducted in Central America, Argentina, Colombia, Uruguay, Chile, and Ecuador, which found working conditions were worse in men than in women [11,29,30]. Another recent study in 13 Latin American countries reported that women were not more vulnerable to the effects of informal employment than men. The authors hypothesized that gender inequalities inherent to the labor market could explain how, even if the increasing risk of poor health in men were related to informal employment, in women, formal employment may not have reached the standards that make it a protective determinant of health. This hypothesis agrees with the findings of other studies [31].

This study has some limitations, mainly related to the data source. Data from the survey in Peru only include urban workers, excluding rural workers, most of whom are agricultural workers. However, the ILO recommends excluding agriculture for measuring informality [32]. Furthermore, the sample was randomly selected and weighted by the distribution of the total working population of Peru. Another limitation is that we assessed health status based on people’s self-perception, which could be affected by cultural and social factors. However, this item has been validated several times, is commonly used in public health studies, and has been demonstrated to be reliable and cost-effective [27]. Finally, as in any cross-sectional study, there is a possible reverse causality bias, which could mean that people with P-SPH could have greater difficulty in finding formal employment. However, this bias is most likely negligible, because the limited supply of formal jobs makes it more likely that both workers with good and P-SPH seek work in the informal economy regardless of their baseline health status. In addition, we did not include the number of hours worked working time of workers, which could lead to differences in the effects on health of poor working conditions, and could be especially different in formal and informal employees.

A strength of our study is that the percentage of informality was similar to those found officially by the ILO [1]. Our results can thus be assumed to be representative of the Peruvian working population. This is the first attempt to assess the association between informal employment and health status by sex considering the role played by working conditions. Finally, we used all available working conditions and employed a novel approach to understand exposure to poor working conditions by calculating a single score that includes all dimensions. This study is a step forward towards better understanding what informal employment supposes for worker’s health and how this effect is partly, but not entirely, driven by working conditions. In a recent editorial [33], the authors discuss the likely interaction between employment and working conditions on the path from informal and precarious employment to decent work. In this way, this study paves the way for future studies to further analyze the impact of poor working and employment conditions (informal employment) on workers’ health.

## 5. Conclusions

Our results suggest that poor working conditions associated with informal employment could partially explain the higher prevalence of P-SPH in informal workers, which was more common in men than in women. This was an expected result; poor working conditions could be a mechanism to explain the effect of informal employment on poor health. Indeed, in light of our results, it could be hypothesized that the transition from informal to formal employment could help to improve working conditions and SPH, since unlike informal jobs, formal jobs, by involving a legal employment relationship, are susceptible to labor inspection, supervision and other control measures [34]. Future studies should analyze other labor, economic and social mechanisms related to informal employment that could further explain this association.

## Figures and Tables

**Figure 1 ijerph-19-06105-f001:**
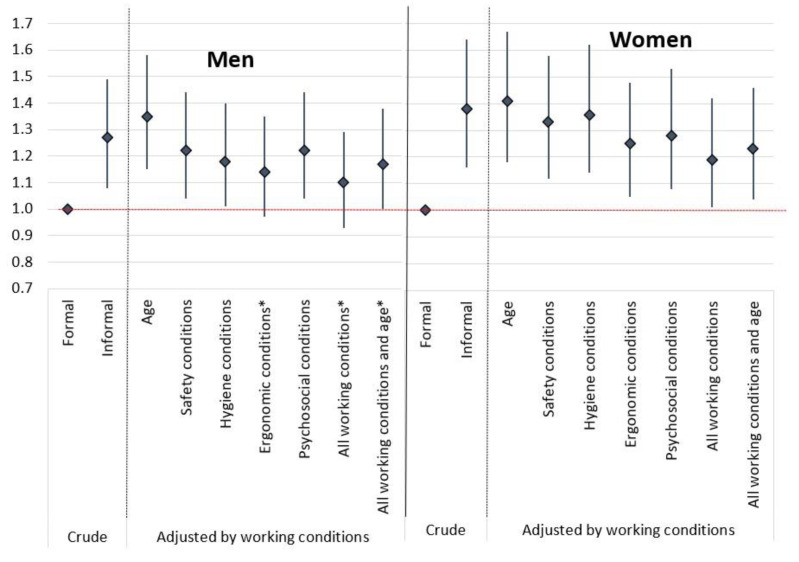
Prevalence ratios (PR) of poor self-perceived health between informal and formal employees (reference category), both crude and adjusted by working conditions and age with 95% confidence intervals (95% CI), working conditions survey of Peru 2017. Working conditions were treated as a continuous variable, and the possible responses in which each category represented a numerical value on the Likert scale (always = 5, many times = 4, sometimes = 3, rarely = 2 and never = 1, except for positive psychosocial conditions phrased positive that scored the other way around) were summed in a score by each working condition dimension (safety, hygiene, ergonomic, psychosocial). * The chi-square statistic is not significant at the 0.05 level.

**Table 1 ijerph-19-06105-t001:** Prevalence of poor working conditions and poor self-perceived health in the working population, stratified by sex and employment status, working conditions survey of Peru 2017.

	Men (*n* = 1621)		*p* Value	Women (*n* = 1477)		*p* Value
	Informal (66.2%)	Formal (33.8%)		Informal (76.6%)	Formal (23.4%)	
**Health Status**						
Poor self-perceived health by age						
≤24	22.7	17.3		29.9	26.3	
25–44	37.7	28.3		46.2	31.5	
45–64	54.9	42.7		66.0	39.0	
≥65	65.7	48.7		87.6	81.8	
All ages	37.6	29.6	<0.05	47.3	34.3	<0.001
**Safety Dimension**						
Fall on the same level	34.5	26.1	<0.001	16.4	13.2	0.2
Fall from height	32.5	26.9	<0.001	14.5	12-0	0.3
Risk of accident with machines or tools	45.4	35.2	<0.001	23.2	12.1	<0.001
**Hygiene Dimension**						
Noise exposure	49.9	40.8	<0.001	26.5	23.7	0.3
Chemical risk exposure	28.5	16.8	<0.001	12.1	8.4	0.1
Dust and fumes exposure	44.8	27.3	<0.001	21.6	15.7	<0.05
Biological risks exposue	8.4	9.5	0.5	8.9	13.7	<0.05
Radiation exposure	53.9	37.3	<0.001	24.4	19.6	0.1
**Ergonomic Dimension**						
Awkward postures	62.7	48.2	<0.001	50.8	36.3	<0.001
Load lifting	63.1	39.4	<0.001	44.3	22.2	<0.001
Repetitive movements	71.7	57.6	<0.001	62.6	53.7	<0.05
**Psychosocial Dimension**						
High work rate	62.6	55.4	<0.05	63.8	61.0	0.4
Low control at work	56.6	64.9	<0.05	57.2	69.8	<0.001
Hiding emotions	41.8	50.0	<0.05	50.7	62.7	<0.001
Not applying knowledge	14.9	8.1	<0.001	20.1	7.4	<0.001
Not learning	18.7	7.8	<0.001	25.0	12.1	<0.001
High workload	36.7	48.8	<0.001	42.7	56.6	<0.001
Lack of supervisor support	55.9	41.7	<0.001	56.1	37.5	<0.001
Lack of coworker support	34.0	21.4	<0.001	41.2	25.0	<0.001
Lack of recognition	31.7	21.5	<0.001	36.5	20.7	<0.001

Working conditions were treated as dichotomous (poor/good). Working conditions were considered as poor in people responding “always”, “often”, “sometimes” and good in those responding “rarely” or “never”. Self-perceived health (SPH) was considered good in workers responding “good” and “very good”, and poor in those responding “fair”, “poor”, or “very poor” health. *p* value: chi-square test.

**Table 2 ijerph-19-06105-t002:** Prevalence of poor self-perceived health according to working conditions by sex, working conditions survey of Peru 2017.

	Men (*n* = 1621)			Women (*n* = 1477)		
	% Poor Self-Perceived Health			% Poor Self-Perceived Health		
	Poor Working Conditions	Good Working Conditions	*p* Value	Poor Working Conditions	Good Working Conditions	*p* Value
**Safety Dimension**						
Fall on the same level	39.06	32.12	0.002	53.77	41.84	<0.001
Fall from height	41.78	31.05	<0.001	56.30	41.75	<0.001
Machines or tools exposure	40.20	30.08	<0.001	58.85	39.80	<0.001
**Hygiene Dimension**						
Noise exposure	36.16	32.71	0.767	48.10	42.17	0.041
Chemical risk exposure	44.67	31.08	<0.001	54.91	42.19	<0.001
Dust and fumes exposure	41.66	29.76	<0.001	52.63	41.49	<0.001
Biological risks exposue	40.59	33.66	0.033	46.62	43.40	0.47018
Radiation exposure	39.01	30.10	0.001	51.93	41.31	0.001
**Ergonomic Dimension**						
Awkward postures	38.88	28.12	<0.001	52.57	35.92	<0.001
Load lifting	38.86	28.93	<0.001	54.13	37.27	<0.001
Repetitive movements	39.33	24.25	<0.001	47.88	37.26	0.001
**Psychosocial Dimension**						
High work rate	33.86	35.22	0.525	42.70	45.58	0.251
Low control at work	32.81	36.60	0.058	43.21	44.48	0.278
Hiding emotions	36.17	32.86	0.977	44.65	42.67	0.707
Not applying knowledge	34.28	33.76	0.673	42.58	49.02	0.085
Not learning	33.06	40.87	0.01	41.65	51.64	0.002
High workload	35.09	32.85	0.304	43.53	43.10	0.537
Lack of supervisor support	28.67	35.41	0.003	36.41	48.00	<0.001
Lack of coworker support	32.08	35.99	0.294	37.50	50.00	<0.001
Lack of recognition	30.28	38.48	<0.001	36.57	53.37	<0.001

Working conditions were treated as dichotomous (poor/good). Working conditions were considered poor in persons responding “always”, “often”, and “sometimes” and good in those responding “rarely” or “never”. Self-perceived health (SPH) was considered good in workers responding “good” and “very good”, and poor in those responding “fair”, “poor”, or “very poor” health. *p* value: chi-square test.

## Data Availability

Data are available on request from the National Institute of Health of Peru and the National Center for Occupational Health and Environmental Health Protection (Spanish acronym: CENSOPAS).

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
