# Peer review of "Informal Employment, Working Conditions, and Self-Perceived Health in 3098 Peruvian Urban Workers"

_ijerph, 2022, doi:10.3390/ijerph19106105_

Round 1

Reviewer 1 Report

The paper „Informal employment, working conditions, and self-perceived health: a cross-sectional study in urban working population in Peru” empirically investigates an important and scientifically interesting question: how much the poorer working conditions are responsible for the worse self-perceived health under informal than under formal employment?

The structure and the wording are clear and understandable, however, there are some grammatical mistakes in the text that should be corrected (like in line 50: impact ‘on health’ and not ‘and health’, or in line 95: there is a duplication of the word ‘by’).

The paper is clearly an empirical work: it practically lacks the theoretical considerations except for a short introduction and a comparison of results with the literature in the discussion section. Basically, it is not a problem, but at least the research model should be grounded. At the same time, it is not a good idea to introduce new literature sources in the discussion (or the conclusion) sections. All those studies should have been mentioned before the materials and methods and could be utilized to establish the empirical research model.

Unfortunately, there are many small problems in the manuscript that should be corrected or answered before publication.

However, because the topic is interesting and important, I hope the authors will do the necessary improvements, and the paper will be published. 

Questions and comments:

  1. Line 44. The priority of formalizing the informal economy is accepted without critique, however, it could be costly, and thus harmful if it is done too fast and too extensively. I am not writing that is a necessarily bad idea, but it should be discussed in more detail.
  2. Line 63. The sample is stated to be representative but is not mentioned to what demographic variables the representativeness is ensured or tested. It is also not mentioned if the representativeness was ensured by applying weights or by what other means. 
  3. Lines 68-74. The dichotomization of some variables was decided, but the reasons are not explained. Why did the authors not use more dummy variables instead to represent more values of the variables? The same problem arises later in lines 84-86 and below tables 1 and 2 as well. It is particularly questionable to assign the medium value of the five-category Likert scales (like ‘sometimes’) to the poor condition. Why these responses are not omitted instead?
  4. Where are the results of the regression analysis mentioned in line 86?
  5. The statistical tests related to Table 1 and Table 2 are unclear. The reader’s only information is that it was an association test, and the actual test type is hidden.
  6. Line 91: how the mentioned “summing-up” was conducted and why that way?
  7. Lines 125-133. Whit what kind of methods the significance of the differences between the two genders was tested? Where is the report of these tests?
  8. The problem of causality is mentioned among the limitations only. This should have been addressed much earlier (among the literature or in the methodological section).
  9. Line 225. Who and why expected these results? 
  10. The authors concluded in lines 227-228 about the formalization of employment as it would be the best way to improve working conditions is not supported by the results of the study at all. In fact, it was not even examined.
  11. Lines 212-214. Why the 70% informality makes the causality bias negligible? It is far from being clear.

Author Response

Reviewer: 2

Comments to the Author:

The paper „Informal employment, working conditions, and self-perceived health: a cross-sectional study in urban working population in Peru” empirically investigates an important and scientifically interesting question: how much the poorer working conditions are responsible for the worse self-perceived health under informal than under formal employment?

The structure and the wording are clear and understandable, however, there are some grammatical mistakes in the text that should be corrected (like in line 50: impact ‘on health’ and not ‘and health’, or in line 95: there is a duplication of the word ‘by’).

The paper is clearly an empirical work: it practically lacks the theoretical considerations except for a short introduction and a comparison of results with the literature in the discussion section. Basically, it is not a problem, but at least the research model should be grounded. At the same time, it is not a good idea to introduce new literature sources in the discussion (or the conclusion) sections. All those studies should have been mentioned before the materials and methods and could be utilized to establish the empirical research model.

Unfortunately, there are many small problems in the manuscript that should be corrected or answered before publication.

However, because the topic is interesting and important, I hope the authors will do the necessary improvements, and the paper will be published.

We greatly appreciate your comments. We have carefully revised the manuscript and have considered all your suggestions. Regarding the grammatical mistakes we have corrected the ones pointed out by the reviewer and the paper has been reviewed by a professional copyediting service. Find the prove attached.

  • Line 44. The priority of formalizing the informal economy is accepted without critique, however, it could be costly, and thus harmful if it is done too fast and too extensively. I am not writing that is a necessarily bad idea, but it should be discussed in more detail

We very much appreciate the suggestions of the reviewer on reflecting on the possible consequences of formalizing the informal economy. The literature shows different point of views regarding this issue. However, all authors agree on the importance of this transition as a way to attain decent work, reduce inequalities among this population and improve their health.

To clarify this in the new version of the manuscript, we have included the following paragraph (Section Introduction, line 51-52):

(…) “These objectives will require significant efforts and the implementation of multiple strategies to facilitate the transition to formality (8).” (…)

  1. International Labour Organization. Transition from the informal to the formal economy-Theory of Change. 2021.

  • Line 63. The sample is stated to be representative but is not mentioned to what demographic variables the representativeness is ensured or tested. It is also not mentioned if the representativeness was ensured by applying weights or by what other means

We appreciate the reviewer concern about the representativeness of the sample. The sampling and data collection was carried out in four stages, 1) selection of clusters (between 100 and 150 defined by the INEI), 2) selection of block in each cluster, 3) selection of nine households in the selected blocks, 4) selection of one person to be interviewed; 2,3,4 stages where carried out in a randomised way, Also for all calculations, the weighting factors of the surveys were applied -which are calculated according to sex, age, economic sector, and geographic area- having the population census of the National Institute of Statistics and Informatics of Peru (INEI by its acronym in Spanish).

In order to clarify how representativeness was achieved, we have included in the text (Section Material and Methods, line 78-79):

(...) "Sample selection was multistage probabilistic with a confidence level of 95%. Survey weighting factors were applied for all calculations.” (…)

And in line 82:

(…) “Detailed description of the survey methodology is available elsewhere (15).” (…)

  1. Sabastizagal-vela I, Astete-Cornejo J, Benavides FG. Condiciones De Trabajo , Económicamente Activa Y Ocupada En in the Economically Active and Employed. Rev Peru Med Exp Salud Publica [Internet]. 2020;37(1):32–41. Available from: http://www.scielo.org.pe/pdf/rins/v37n1/1726-4642-rins-37-01-32.pdf

  • Lines 68-74. The dichotomization of some variables was decided, but the reasons are not explained. Why did the authors not use more dummy variables instead to represent more values of the variables? The same problem arises later in lines 84-86 and below tables 1 and 2 as well. It is particularly questionable to assign the medium value of the five-category Likert scales (like ‘sometimes’) to the poor condition. Why these responses are not omitted instead?

We thank the reviewer’s reflection on the treatment of the variable SPH as dichotomous. Self-perceived health (SPH) has increasingly been used to measure health status since this measure has demonstrated to be reliable, valid, simple, and cost-effective. In most of the studies the five-point Likert scale is dichotomized considering as poor SPH those who reported fair, poor, and very poor health (we included some examples below).

Maheswaran, H., Kupek, E., & Petrou, S. (2015). Self-reported health and socio-economic inequalities in England, 1996-2009: Repeated national cross-sectional study. Social Science and Medicine, 136–137. https://doi.org/10.1016/j.socscimed.2015.05.026

Zajacova, A., Huzurbazar, S., & Todd, M. (2017). Gender and the structure of self-rated health across the adult life span. Social Science and Medicine, 187, 58–66. https://doi.org/10.1016/j.socscimed.2017.06.019

Mossey, J. M., & Shapiro, E. (1982). Self-Rated Health: A Predictor of Mortality Among the Elderly. https://www.ncbi.nlm.nih.gov/pmc/articles/PMC1650365/pdf/amjph00655-0034.pdf

Jürges, H., Avendano, M., & MacKenbach, J. P. (2008). Are different measures of self-rated health comparable? An assessment in five European countries. European Journal of Epidemiology, 23(12), 773–781. https://doi.org/10.1007/s10654-008-9287-6

Also, to clarify this point we have added to the manuscript some examples of studies that use the variable in the same way (Section Material and Methods, line 87):

“…which was dichotomized as is usual in most epidemiological studies (16)...”

  1. Jürges H, Avendano M, MacKenbach JP. Are different measures of self-rated health comparable? An assessment in five European countries. European Journal of Epidemiology. 2008;23(12):773–81.

Regarding the consideration of workers who report “sometimes” to working condition questions as having poor working conditions, we agree with the reviewer that it could be considered that exposure is overestimated. However, there is a big body of literature that defines exposition the same way as we did on this study. Hence, for comparison purposes we decided to proceed in the same way. We have also added a new sentence and reference as follows (Section Material and Methods, line 102-103):

(…) “This is the most common way of analysing exposure to poor working conditions in a description analysis (17).” (…)

  1. Nappo N. Is there an association between working conditions and health? An analysis of the Sixth European Working Conditions Survey data. 2019; Available from: https://doi.org/10.1371/journal.pone.0211294

  • Where are the results of the regression analysis mentioned in line 86?

We thank the reviewer for pointing out the lack of clarity on where the regression analysis’ results are shown. We have clarified next to the comment the table in which the results are shown (Section Material and Methods, line 106):

(…)” For regression analysis, exposure to working conditions was treated as a continuous variable (Table 3).” (…)

  • The statistical tests related to Table 1 and Table 2 are unclear. The reader’s only information is that it was an association test, and the actual test type is hidden

We thank this comment. We have used a Chi-square to test assess differences between formal and informal employees across descriptive variable categories.

Accordingly, we have added this information to table footnotes and the following to the manuscript (Section Material and Methods, line 118-119):

(…) “Chi-square test was calculated to assess differences in the distribution between comparison groups.” (…)

  • Line 91: how the mentioned “summing-up” was conducted and why that way?

We appreciate the reviewer’s comment on the treatment of the working conditions on the regression model. We considered several items on each dimension of working conditions. Hence, following the principle of parsimony we carried out the analysis taking all items into account individually and by constructing a score for each dimension (see example below). We obtained similar results and following the principle of parsimony we decided to treat each dimension (ergonomic, hygienic, psychosocial, safety) as a score made up by summing up responses from the Likert scale obtained in each item.

For the safety dimension there were 3 items: Fall on the same level, fall from height, risk of accident with machines or tools. If worker X had answered on the Likert scale “sometimes=3”, “always=5”, and “rarely=2”, the score of this worker for safety dimension would be 10.

  • Lines 125-133. Whit what kind of methods the significance of the differences between the two genders was tested? Where is the report of these tests?

Thanks for the comment. In our study all the calculations were stratified by sex, however, as it was not the aim of this study we didn’t measure sex differences. However, if the Editore considers it, we can formally make this comparison.

  • The problem of causality is mentioned among the limitations only. This should have been addressed much earlier (among the literature or in the methodological section).

We thank the reviewer the suggestion, we have addressed this limitation on the Section Materials and Methods, line 121-124 of the manuscript:

(…) “PR were interpreted cautiously due to the cross-sectional nature of the study, which could induce a reverse causality bias.” (…)

  • The authors concluded in lines 227-228 about the formalization of employment as it would be the best way to improve working conditions is not supported by the results of the study at all. In fact, it was not even examined.

We appreciate the reviewer's invitation to ponder our conclusion. Our results show how formal workers have a lower prevalence of P-SPH and exposure to poor working conditions (safety, hygiene and ergonomics) than informal ones. Also, we found that higher prevalence of P-SPH also coincides with higher prevalence of exposure to poor working conditions. This led us to consider that the transition from informality to formality would mean an improvement in working conditions, and in consequence in self-perceived health. Nevertheless, it is true that this hypothesis can´t be proved on light of our results, nor with a cross-sectional study. We have modified the conclusions accordingly (Section Conclusions, lines 260-261):

(…) “. Indeed, in light of our results, it could be hypothesized that the transition from informal to formal employment could help to improve working conditions and SPH” (…)

  • Lines 212-214. Why the 70% informality makes the causality bias negligible? It is far from being clear.

We thank the reviewer for pointing out the lack of clarity regarding causality bias. Causality bias is universal in cross-sectional studies because, as it is well known, we can´t be sure about the temporality criteria, whether informal employees have poor-SPH due to the precarity they are exposed to (poor working conditions) or if workers with poor-SPH are more prone to look for informal jobs. However, in a country like Peru, we believed that this bias cold be almost negligible because most of its labour market is informal, and thus both workers with good and poor-SPH have more access to an informal job. Following the reviewer’s suggestion, we have clarified this point (Section Discussion, lines 240-242 as follows:

“…this bias is most likely negligible due to the fact that the limited offer of formal jobs makes more likely that both workers with good and poor-SPH is into informal economy regardless of their baseline health status.”(…)

Reviewer 2 Report

Informal employment, working conditions, and self-perceived health: a cross-sectional study in urban working population in Peru

IJERPH 1634692 Peer Review Report

This article brings to light interesting descriptive statistics. Its strength lies in laying out rates of informal employment as well as exposure to various workplace hazards by gender. I lay out some questions and suggestions below:

  • The writing in this manuscript needs significant editing. There is some inappropriate formality (e.g. “probably” in the opening sentence), as well as some incorrect word choices (e.g. “separated by” instead of “separately for” in the abstract). There are also examples of awkward sentence structure, incorrect punctuation, and misspelled words.
  • The authors never define informal work, which makes the opening paragraph (and much of the article) confusing to most readers.
  • ILO is not defined
  • The authors should do a better job justifying why they examine difference between men and women. That is, is there research or theory that suggests such differences are likely to exist? What mechanisms are at play?
  • The authors should explain why the survey question they use is appropriate for approximating informal work.
  • The authors should also spend some time justifying their dependent variable. What might they be missing by using a question that only asks about the last two weeks, especially among those who worked very little or were on vacation at the time of the survey?
  • The authors should probably control for amount of work time, as this is likely to explain a lot of the variance in exposure to particular work conditions and also health outcomes
  • Why did the authors use Poisson models?
  • Why did the authors use both Stata and SPSS?
  • The authors explain that correlates of informal work are likely what explain differences in health; what, then, can the authors actually say in this paper about the role of informal work itself, beyond that it correlates with better predictors of poor health?

Author Response

Comments to the Author:

This article brings to light interesting descriptive statistics. Its strength lies in laying out rates of informal employment as well as exposure to various workplace hazards by gender. I lay out some questions and suggestions below:

The writing in this manuscript needs significant editing. There is some inappropriate formality (e.g. “probably” in the opening sentence), as well as some incorrect word choices (e.g. “separated by” instead of “separately for” in the abstract). There are also examples of awkward sentence structure, incorrect punctuation, and misspelled words

We greatly appreciate your comments which we have addressed below. Regarding the language errors, the paper has been reviewed by a professional copyediting service. Find the prove attached

  • The authors never define informal work, which makes the opening paragraph (and much of the article) confusing to most readers.

We appreciate the suggestion of clarifying informality concept. The ILO has defined the Informal Economy as “all the economic activities of workers and economic units that – in law or in practice – are not covered or are insufficiently covered by formal systems” Nevertheless, there is no consensus on the definition of informal employment. In order to guide the countries to have indicators, in the ILO’s 17th International Conference of Labour Statisticians, informal employment was defined as the total number of informal jobs carried out in: i) informal sector companies, ii) in formal sector companies or iii) households.

Following this consensual definition, the text we will include (Section Introduction, line 40-45):

(…) The international Labour Organization (ILO) has defined informal employment as the total number of informal jobs whether in formal sector enterprises, in informal sector enterprises, or as paid domestic workers by households. Informal jobs are those whose employment relationship is not subject to national labour legislation, income taxation, social protection or entitlement to employment benefits (…)

  1. International Labour Organization. Informality and non-standard forms of employment. Buenos Aires; 2018.

  • ILO is not defined

Thanks for noticing the absence of ILO’s definition, we have included it on the text as suggested in Section Introduction, line 40:

 “… the International Labour Organization (ILO) has …”

  • The authors should do a better job justifying why they examine difference between men and women. That is, is there research or theory that suggests such differences are likely to exist? What mechanisms are at play?

We appreciate your suggestion. We have incorporated a new paragraph (Section Introduction, lines 59-63).

(…) “In addition, more and more women are entering the labor market, yet they continue to be placed in greater proportion in more precarious jobs, with lower wages, high levels of informality, and greater job insecurity. This, together with women's unequal participation in unpaid work driven by gender inequalities, can have a detrimental effect on their general and mental health (13).”  (...)

  1. Jung AK, O’Brien KM. The Profound Influence of Unpaid Work on Women’s Lives: An Overview and Future Directions. Journal of Career Development. 2019;46(2):184–200.

  • The authors should explain why the survey question they use is appropriate for approximating informal work.

We appreciate and agree that it is important to justify the criteria we take to approach informal work. Firstly, and as the reviewer suggested in comment #1 we added the ILO’s definition of informal employment as follows (Section Introduction, Line 40):

(...) The international Labour Organization (ILO) has defined informal employment as the total number of informal jobs whether in formal sector enterprises, in informal sector enterprises, or as paid domestic workers by households. Informal jobs are those whose employment relationship is not subject to national labour legislation, income taxation, social protection or entitlement to employment benefits (…)

Secondly, we added an explanation of how informal employment, as defined by ILO, could operationalized in Peru, which is by affiliation to a Pension System, a requirement to be in formal employment relationships (Section Materials and Methods, Line 84-87)

(…) “We have defined informal employment by the lack of a contract and of social security coverage. In Peru, as in many countries, legally recognized workers must be affiliated to a Pension System. Thus, not being affiliated is a close proxy to informal employment.” (…)

  • The authors should also spend some time justifying their dependent variable. What might they be missing by using a question that only asks about the last two weeks, especially among those who worked very little or were on vacation at the time of the survey?

Thank you for the comment. The inclusion criteria used in the survey was based in the ILO definition of being employed. That is defined as:

 “All those of working age who, during a short reference period, were engaged in any activity to produce goods or provide services for pay or profit. They comprise employed persons “at work”, i.e. who worked in a job for at least one hour; and employed persons “not at work” due to temporary absence from a job, or to working-time arrangements (such as shift work, flextime and compensatory leave for overtime). “ 

This definition contemplates any reason for being temporary absent from work, which could include being on vacation, illness or leave. Also, this criterion is being used in the literature before, so its use helps comparison between studies.

We have added a new sentence in the manuscript and a reference to the manuscript in order to justify our decision (Section Materials and Methods, line 77):

(…) “Which, according to the ILO, is the defining characteristic of being an employee (14).” (…)

  1. International Labour Organization. Concepts and definitions - ILOSTAT [Internet]. 2021 [cited 2022 Apr 12]. Available from: https://ilostat.ilo.org/resources/concepts-and-definitions/

  • The authors should probably control for amount of work time, as this is likely to explain a lot of the variance in exposure to particular work conditions and also health outcomes

Thank you for your appreciation. We believe working hours are important employment condition However it is not in the scope of our study since our objective and scope were limited to working conditions. In fact, we have considered that into limitations and will be considered in future studies (Section Discussion, lines 243-245):

(…) “In addition, we did not include the number of hours worked working time of workers, which could lead to differences in the effects on health of poor working conditions, and could be especially different in formal and informal employees..” (…)

  • Why did the authors use Poisson models?

Thank you for the comment. The use of Poisson regression models with robust variance to estimate prevalence ratios with a binary dependent variable are widely proved, it has been proved to be preferable to other regression models because it provides unbiased estimates, and it is commonly used in public health literature.

We have added a sentence in Material and Methods Section justifying the use of the model (line 121-122):

(…) “This model was selected over other regression models because it provides unbiased estimates(18, 19).” (…)

  1. Zou GY, Donner A. Extension of the modified Poisson regression model to prospective studies with correlated binary data. Stat Methods Med Res [Internet]. 2013;22(6):661–70.

  1. Chen, W., Qian, L., Shi, J. et al. Comparing performance between log-binomial and robust Poisson regression models for estimating risk ratios under model misspecification. BMC Med Res Methodol 18, 63 (2018). https://doi.org/10.1186/s12874-018-0519-5

  • Why did the authors use both Stata and SPSS?

We thank the reviewer’s comment, and agree that this can be confusing. At the beginning we used both softwares, but finally we conducted all analyses in Stata. Therefore, modified the text accordingly (Section Material and Methods, line 126).

(…) “The analyses were conducted in Stata v.13.”

  • The authors explain that correlates of informal work are likely what explain differences in health; what, then, can the authors actually say in this paper about the role of informal work itself, beyond that it correlates with better predictors of poor health?

We appreciate the reviewer’s question. We agree that it is worth explaining better the possible effect related to informality. Generally, entrance of people into the informal economy is not by choice. Most of workers who enter into this unregulated employment agreements cannot find a job or start businesses in the formal economy, especially in a context like the Peruvian, with such high informality rates. Informal work, in addition to being related to the P-SPH, also represents a social and economic problem for the country. Informal workers are unprotected, they don’t have a system that guarantees them a decent retirement, nor do they have health protection. In addition, companies in an informal situation don’t pay taxes, which affects the amount resources used to improve state services. For example, informality played an unfortunate role during the COVID 19 pandemic, preventing economic aid designed by the state from reaching people who were not registered. Hence, informal employment plays a complex role in the countries development that ultimately touches people’s health by affecting from distal to proximal determinants of populations health.

We have further addressed this question on the manuscript (Section Discussion, Line 189-193):

(…) “However, some of the effect could be explained by informality per se, which could be re-lated to other mechanisms such as income insecurity, poverty, precarious living condi-tions, lack of access to health services and the absence of social protection benefits. These social and economic factors, together with living conditions, could play an important role in this association. (…)

Reviewer 3 Report

  • Please extend your analysis to examine the relationship between the prevalence of poor working conditions and perception of one's own health and the age of the respondents (breakdown by gender and age groups - can be ranges, 5-10 year ranges).
  • Please introduce a table with a breakdown by gender and age groups, with the size of each group shown. It seems that older people will have a higher rate of ill health.
  • Please see if age is correlated with perception of own health.
  • In the introduction, it would be useful to refer to other similar studies already carried out.
  • Are there plans to expand the study in the future?
  • Please consider replacing some table with a graph, or provide an additional graph.
  • Please consider using a different survey method in the future.

Author Response

Reviewer: 1

  • Please extend your analysis to examine the relationship between the prevalence of poor working conditions and perception of one's own health and the age of the respondents (breakdown by gender and age groups - can be ranges, 5-10 year ranges).

We thank the reviewer the suggestion. We agree that age plays an important role in this association. In this regard, we adjusted the regression for age to control for possible differences in SPH by age. In addition, in the new version of this manuscript, we have added a description of the sample’s age distribution in Table 1 to assess self-perceived health (SPH) by age group. Please, see below:

Men (N=1621)

p value

Women (N=1477)

p value

Informal (66.2%)

Formal (33.8%)

Informal (76.6%)

Formal (23.4%)

Health status

Poor self-perceived health by age

<0.05

<0.001

<=24

22,7

17,3

29,9

26,3

25 - 44

37,7

28,3

46,2

31,5

45 - 64

54,9

42,7

66,0

39,0

>=65

65,7

48,7

87,6

81,8

All ages

37.6

29.6

47.3

34.3

  • Please introduce a table with a breakdown by gender and age groups, with the size of each group shown. It seems that older people will have a higher rate of ill health.

We thank the reviewer for pointing out the importance of showing the age distribution of the sample in the manuscript. We have included a table with a breakdown by sex and employment status of age groups. As shown below:

 Supplementary table 1. Distribution of the working population by age stratified by sex and employment status, Peru’s Working Conditions Survey 2017.

We added a consideration regarding difference in age by employment status in Section Methods, line 79-80:

(…) “Informal workers of both sexes were younger than formal workers (Supplementary Table 1). (…)

  • Please see if age is correlated with perception of own health

We appreciate the opportunity to explore and clarify the probable correlation between age and SPH. As stated above we have carried out a descriptive analysis to assess this relationship in table 1, where we can see that a significant relationship exists and, as expected, older workers report higher prevalence of poor SPH. We considered this correlation in Poisson regression models by adjusting by age.

  • In the introduction, it would be useful to refer to other similar studies already carried out

We agree with the reviewer. The manuscript would benefit from more references to justify this study. As space is limited, we have synthesized the most important information by citing some similar studies:

  1. Rodriguez-Loureiro L, Vives A, Martínez Franzoni J, Lopez-Ruiz M. Health inequalities related to informal employment: gender and welfare state variations in the Central American region. Critical Public Health [Internet]. 2020;30(3):306–18. Available from: https://doi.org/10.1080/09581596.2018.1559923
  2. Jung AK, O’Brien KM. The Profound Influence of Unpaid Work on Women’s Lives: An Overview and Future Directions. Journal of Career Development. 2019;46(2):184–200.
  3. Sabastizagal-vela I, Astete-Cornejo J, Benavides FG. Condiciones De Trabajo, Económicamente Activa Y Ocupada En in the Economically Active and Employed. Rev Peru Med Exp Salud Publica [Internet]. 2020;37(1):32–41. Available from: http://www.scielo.org.pe/pdf/rins/v37n1/1726-4642-rins-37-01-32.pdf.
  4. Giatti L, Barreto SM, César CC. Informal work, unemployment and health in Brazilian metropolitan areas, 1998 and 2003. Cadernos de Saude Publica. 2008;24(10):2396–406.
  5. Silva-Peñaherrera M, Lopez-Ruiz M, Merino-Salazar P, Gómez-García AR, Benavides FG. Health inequity in workers of Latin America and the Caribbean. International Journal for Equity in Health [Internet]. 2020 Dec 1;19(1):109. Available from: https://equityhealthj.biomedcentral.com/articles/10.1186/s12939-020-01228-x

However, we have taken the opportunity to include a few more references:

  1. Ruiz ME, Vives A, Martínez-Solanas È, Julià M, Benach J. How does informal employment impact population health? Lessons from the Chilean employment conditions survey. Safety Science. 2017 Dec 1;100:57–65.

  1. Giatti L, Barreto SM, César CC. Informal work, unemployment and health in Brazilian metropolitan areas, 1998 and 2003. Cadernos de Saude Publica. 2008;24(10):2396–406.

  • Are there plans to expand the study in the future?

Thank you for asking. This study is part of a broad project that includes occupational health surveillance in Latin American and Caribbean (LAC) countries. Our research team is attempting to collect the best occupational health information available in Peru as well as in other LAC countries to go further in analyzing the impact of poor working conditions and employment (informal employment) on workers' health. In consequence, we have added a brief sentence in the last paragraph of Discussion section line 25.

(…) “This study opens the door to future studies to further analyze the impact of poor working and employment conditions (informal employment) on workers' health.” (…) 

  • Please consider replacing some table with a graph, or provide an additional graph

We thank the reviewer the suggestion of adding graphs to make the interpretation of the analyses lighter. We have added figure 1 (see below) in place of table 3.

  • Please consider using a different survey method in the future

The reviewer has pointed out a very important issue of the research area which is the lack of readily available data sources. This survey constitutes the first and most recent, reliable data in working and employment conditions and occupational health in Peru, although it has several limitations. Nevertheless, we hope that our research contributes to highlight the imperative necessity of collecting better data that is essential for monitoring worker’s health. 

Round 2

Reviewer 3 Report

I accept all remarks.

Author Response

No attachment found, we reply to academic editor in the next page